# Exome Sequencing Reveals *SLC4A11* Variant Underlying Congenital Hereditary Endothelial Dystrophy (CHED2) Misdiagnosed as Congenital Glaucoma

**DOI:** 10.3390/genes14020310

**Published:** 2023-01-25

**Authors:** Khazeema Yousaf, Sadaf Naz, Asma Mushtaq, Elizabeth Wohler, Nara Sobreira, Bo-Man Ho, Li-Jia Chen, Wai-Kit Chu, Rasheeda Bashir

**Affiliations:** 1Department of Biotechnology, Lahore College for Women University, Lahore 54000, Pakistan; 2School of Biological Sciences, University of the Punjab, Quaid-i-Azam Campus, Lahore 54590, Pakistan; 3Department of Ophthalmology, Children’s Hospital & the Institute of Child Health, Lahore 54000, Pakistan; 4Hong Kong Hub of Paediatric Excellence, The Chinese University of Hong Kong, Hong Kong 999077, China; 5McKusick-Nathans Department of Genetic Medicine, Baylor Hopkins Center for Mendelian Genomics, Baltimore, MD 21205, USA; 6Department of Ophthalmology and Visual Sciences, The Chinese University of Hong Kong, Hong Kong 999077, China

**Keywords:** congenital hereditary endothelial dystrophy, primary congenital glaucoma, intraocular pressure

## Abstract

Autosomal recessive congenital hereditary endothelial dystrophy (CHED2) may be misdiagnosed as primary congenital glaucoma (PCG) due to similar clinical phenotypes during early infancy. In this study, we identified a family with CHED2, which was previously misdiagnosed as having PCG, and followed up for 9 years. Linkage analysis was first completed in eight PCG-affected families, followed by whole-exome sequencing (WES) in family PKGM3. The following in silico tools were used to predict the pathogenic effects of identified variants: I-Mutant 2.0, SIFT, Polyphen-2, PROVEAN, mutation taster and PhD-SNP. After detecting an *SLC4A11* variant in one family, detailed ophthalmic examinations were performed again to confirm the diagnosis. Six out of eight families had *CYP1B1* gene variants responsible for PCG. However, in family PKGM3, no variants in the known PCG genes were identified. WES identified a homozygous missense variant c.2024A>C, p.(Glu675Ala) in *SLC4A11*. Based on the WES findings, the affected individuals underwent detailed ophthalmic examinations and were re-diagnosed with CHED2 leading to secondary glaucoma. Our results expand the genetic spectrum of CHED2. This is the first report from Pakistan of a Glu675Ala variant with CHED2 leading to secondary glaucoma. The p.Glu675Ala variant is likely a founder mutation in the Pakistani population. Our findings suggest that genome-wide neonatal screening is worthwhile to avoid the misdiagnosis of phenotypically similar diseases such as CHED2 and PCG.

## 1. Background

Primary congenital glaucoma (PCG; OMIM 231300) is a severe form of optic neuropathy which is characterized by an abnormal development of the trabecular meshwork and high intraocular pressure (IOP) at birth or within the first 3 years of life, resulting in severe tearing, buphthalmos (enlarged eyeballs) and photophobia [1]. PCG has an autosomal recessive pattern of inheritance with variable penetrance. [2] Variants in *CYP1B1*, *LTBP2*, *MYOC*, *NTF4*, *FOXC1* and *WDR36* have been reported to cause PCG. [3] *CYP1B1* gene variants are the major contributors to PCG [1]. 

Congenital hereditary endothelial dystrophy (CHED) is a rare genetic disorder involving corneal endothelium and has been classified into two forms—CHED1, the autosomal dominant form (OMIM 121700) and CHED2, (OMIM 217700) the autosomal recessive form. [4] Loci for CHED1 and CHED2 were mapped to chromosome 20p11.2-q11.2 [5] and 20p13 [6], respectively. CHED2 is typically seen at birth or during the early years of life [7]. CHED is characterized by symmetrical, noninflammatory and bilateral corneal clouding (edema) that causes decreased vision [7]. Degeneration of the Descemet membrane (DM) and corneal endothelium leads to diffuse ground-glass-like corneal edema; usually, there is no tearing or photophobia in CHED patients as only fine corneal epithelial edema and no bullae are found with reduced vision and sometimes rotatory nystagmus [7]. In CHED patients, a normal corneal thickness and endothelium without epithelial edema and a thickened DM can be detected [8]. On the other hand, patients with PCG usually present with tearing, photophobia, inflammation, high IOP with optic nerve damage and breaks in the DM [8].

Variants of the human sodium bicarbonate transporter-like solute carrier family 4 member 11 *SLC4A11* (NM_032034.3) gene cause CHED2 in different populations, including Pakistan [7,9]. The *SLC4A11* gene is located on chromosome 20p12 [10]. Variants of *SLC4A11* are not only associated with CHED2 but are also causative of a rare syndromic form of CHED2, which includes deafness as a presenting feature in Harboyan syndrome [11]. Pathogenic variants of *SLC4A11* lead to loss of function in two ways; one is by preventing membrane targeting, and the second is by nonsense-mediated mRNA decay [12]. Of the more than 85 reported *SLC4A11* variants, 57 have been associated with CHED [12]. Most variants in *SLC4A11* have been reported in CHED patients from India [13,14]. However, in four different studies from India, no variant was identified in 4 of 16 (25%), 7 of 42 (17%), 11 of 20 (55%) and 9 of 20 (45%) CHED families [13,14,15,16]. Another report from Bosnia also did not find any *SLC4A11* variants in CHED patients, suggesting a genetic heterogeneity of CHED [17]. Recently, a variant in the Multiple PDZ Domain Crumbs Cell Polarity Complex Component (*MPDZ*) gene was identified in an Iranian CHED patient [17].

There are 19 exons in *SLC4A11*, which encodes a protein containing 891 amino acids with a molecular mass of 100 kDa [18]. SLC4A11 consists of 14 transmembrane domains (TMDs) [10]. According to the NCBI Unigene expression profile, *SLC4A11* is expressed in many human organs and tissues including the eye. *SLC4A11* exists as a dimer [12]. Heteromeric interactions could occur between wild-type (WT) and mutant proteins [12]. SLC4A11 is a sodium-borate co-transporter protein that initiates cell growth by increasing the number of borate ions and activating the mitogen-activated protein kinase (MAPK) pathway [18]. To date, there have been two reports from Pakistani CHED2 patients presenting three different variants, i.e., p.Ser489Leu, p.Gly464Asp [19] and p.Glu675Ala [20].

Although there are detailed criteria for the clinical diagnosis of PCG, certain conditions are known to have overlapping phenotypes with other ocular diseases and can lead to misdiagnosis [8]. It has been previously reported that *SLC4A11* mutations causing CHED2 may be confused with PCG caused by *CYP1B1* variants [20].

In the current study, we conducted a variant analysis of multiple PCG families from Pakistan and identified a family that was misdiagnosed with PCG who had CHED2, which was confirmed after the identification of a variant in *SLC4A11* and clinical reassessment.

## 2. Methods

### 2.1. Ethical Approval and Clinical Testing

This study was approved by the Institutional Review Board of Lahore College for Women University; Children Hospital Lahore, the Institute of Child Health; Mayo Hospital; and School of Biological Sciences, University of the Punjab. Eight families were enrolled in the study from Children Hospital, the Institute of Child Health and the Mayo Hospital, Lahore. The selected participants had consanguineous parents. The patients exhibited no other detectable phenotype except for visual impairment. Written informed consent was obtained from participants and their legal guardians.

Clinical details and linkage analysis with genetic findings for seven families, namely PKH1, PKH2, PKH3, PKH4, PKGM1, PKGM2, and HPK1, were presented previously [21,22]. Among the diagnosed PCG families, seven members in one family, PKGM3, experienced onset of PCG 9 years ago. In the present study, the patients and their unaffected family members underwent detailed ophthalmic evaluations, including slit lamp bio-microscopy, indirect ophthalmoscopy, and corneal pachymetry by pediatric ophthalmologists. An updated clinical diagnosis of CHED leading to secondary glaucoma was made. The presence of other dystrophies, including congenital hereditary stromal dystrophy, posterior polymorphous corneal dystrophy, and congenital glaucoma were excluded. 

### 2.2. Sampling and Whole-Exome Sequencing 

From the participants, 5 mL of peripheral blood was collected in EDTA vials (BD, USA) and the DNA was extracted by performing sucrose lysis and the salting-out method. Samples from individuals III:3, IV:1 and IV:5 in the family PKGM3 were subjected to whole-exome sequencing, which was performed at the Baylor–Hopkins Center for Mendelian Genomics (BHCMG). The Agilent SureSelect Human All ExonV5 kit was used for exome capturing and a low input library preparation protocol was implemented [23]. Libraries were sequenced on the Illumina HiSeq2500 platform to generate 125 bp paired-end reads; variant calling was completed using GATK 3.3-0 joint calling with HaplotypeCaller. The output data were analyzed at BHCMG and the Department of Biotechnology, Lahore College for Women University, Lahore. To annotate the variant call files (VCF), the online program wANNOVAR (http://wannovar.usc.edu/) was used (accessed on 2 January 2022). Output data obtained from wANNOVAR were filtered and compared to the frequencies of the population in the 1000 Genomes database, Genome Aggregation Database (gnomAD) and the Exome Aggregation Consortium (ExAC) database (accessed on 2 January 2022. Variants with an allelic frequency of less than 0.01 were evaluated further. Hemizygous, homozygous, splice-site, and compound heterozygous exonic variants were examined [24]. For further analyses, the individual test exome was compared to a matched set of samples [25]. CNV calls were annotated using AnnotSV. Candidate CNVs were prioritized according to minor allele frequency, exon number, Bayes factor (BF), and the ratio of the observed/expected number of reads [24]. The wANNOVAR files also contained predicted pathogenic scores for the identified variants from MutationTaster; Polyphen 2 and SIFT also had pathogenicity scores of CADD, which indicates the impact of the variant on the function of the protein. 

### 2.3. Sanger Sequencing

After WES analysis, a missense variant c.2024A>C, p.Glu675Ala in *SLC4A11* was identified to be associated with CHED2 in the PKGM3 family. To confirm the segregation pattern of this variant, exon 15 of *SLC4A11* was amplified with polymerase chain reaction (PCR), followed by direct sequencing performed on an ABI 3100 sequencer with Big Dye Terminator® (V 3.1) Cycle Sequencing Kit (Applied Biosystems, Foster City, CA, USA). 

### 2.4. In-Silico Analysis

Clustal Omega was used for multiple alignments of the SLC4A11 protein of different species to check the conservation of the Glu675 residue (Figure 1I) (http://www.ebi.ac.uk/Tool/msa/clustalo) (accessed on 1 February 2022). The damaging effects on protein structure were predicted with Polyphen, PROVEAN and MutationTaster. To investigate the damaging effects of mutation on the stability of proteins, I-Mutant 2.0 and PhD-SNP tools were used. 

## 3. Results

### 3.1. Clinical Features in Members from the PKGM3 Family

Seven members from the PKGM3 family originally diagnosed with PCG were identified (Table 1). This family is from a village in Punjab. Multiple consanguineous marriages were identified in this family. However, only six affected individuals were enrolled in this study because individual IV:4 passed away suddenly during the study period for an unknown reason. Six non-PCG individuals were also enrolled. All the patients were born in the fourth generation of the pedigree (Figure 1A). A detailed medical history was obtained. Ophthalmological examinations were completed at the Children Hospital and the Institute of Child Health hospital in Lahore. Patients from this family had been initially diagnosed with PCG for 9 years. However, a later re-evaluation found that these six members were affected with CHED2, leading to secondary glaucoma. 

In the first sub-family pedigree, individuals IV:1, IV:2 and IV: 3 (Table 1) in the fourth generation had bilateral cloudy cornea (Figure 1B–D). The ages ranged from 8 to16 years. The optic disc cup-disc ratios ranged from 0.3 to 0.4 mm. The IOPs ranged from 18 to 21 mmHg for the three affected individuals. Patients IV:1 and IV:2 both had nystagmus. Visual acuity was reduced to light perception and confined to counting fingers for individuals IV:1 and IV:2, respectively. Individual IV:1 underwent bilateral trabeculectomy (glaucoma surgery) at the age of 6 years. Individual IV:1 also had a bilateral corneal transplant at the age of 12 years. However, after 1 year, bilateral graft rejection was reported. For individual IV:2, folds in the Descemet membrane and bilateral flat bleb were observed. This patient developed nystagmus later on. This patient had undergone a trabeculectomy in the right eye. At the age of 8 years, the right eye had no vision and the left eye had a hazy view.

Individual IV:3 was a 6-year-old boy with symptoms of a hazy cornea (Figure 1D). This patient was misdiagnosed as PCG for 5 years. A later re-evaluation found that he suffered from CHED2, leading to secondary glaucoma. The corneal diameter was 23.5/26.2 (OD/OS). A corneal diameter of 12/12 mm (OD/OS) was recorded. Visual acuity was evaluated by counting fingers for both eyes. As the chance of graft rejection could be high in infancy, penetrating keratoplasty was not performed in this patient. The patient was using medication to control IOP. 

In the second sub-family pedigree, individuals IV:6 and IV:7 in the fourth generation, aged 16 and 14 years old, respectively, were also initially diagnosed with PCG, which later was re-evaluated as CHED2 leading to secondary glaucoma (Figure 1B–D). Both individuals had bilateral cloudy corneas and blurred vision. The optic disc cup-disc ratios ranged from 0.3 to 0.4 mm. The IOPs ranged from 19 to 21.1mmHg. Corneal diameters ranged from 11 to 15mm for the two affected individuals. Visual acuity was reduced to hand movement in the right eye and counting fingers in the left eye in IV:6, while no perception of light in both eyes was detected in individual IV:7. 

Since the initial diagnosis of PCG in IV:6, a bilateral trabeculectomy had been performed. Additionally, penetrating keratoplasty was performed in the right eye at the age of 5 years. However, the corneal graft was not able to improve vision. Individual IV:7 had a bilateral trabeculectomy and bilateral penetrating keratoplasty. Unfortunately, both procedures did not result in any improvement. 

In the third sub-family pedigree, proband IV:9 in the fourth generation was enrolled. Proband IV:9 was a 19-year-old male who was first examined at the age of 4 months with symptoms of corneal haze (Figure 1F). This patient reported some level of photophobia in both eyes. The diagnosis was also PCG. At the age of 6 years, the re-evaluation diagnosis was CHED2. Visual acuity was recorded as no perception of light in both eyes. A Cup-disc ratio of 0.5/0.6 (OD/OS), an axial length of 23.45/24.53 (OD/OS) and an IOP of 22/21 mmHg (OD/OS) were recorded. The patient had undergone a bilateral trabeculectomy. Five years after the bilateral trabeculectomy, the patient underwent bilateral keratoplasty and later reported bilateral graft rejection. Currently, the patient has no vision. 

### 3.2. Genetic Results after Whole-Exome Sequencing

Whole-exome sequencing identified a causative pathogenic missense variant c.2024A>C, p.Glu657Ala (Figure 1A,B) in exon 15 of the *SLC4A11* gene in all six members affected with CHED2 and secondary glaucoma. Sanger sequencing of exon 15 of *SLC4A11* confirmed the segregation of the variant with the phenotype (Figure 1G,H). The parents of the patients were heterozygous and phenotypically normal. Thus, the dominant mode of inheritance, which would result in CHED1, was excluded. The identified variant was screened in 100 ethnically matched controls and this variant was not identified in any control sample.

### 3.3. In Silico Analyses Predict the Variant to Be Damaging to Protein

The p.Gly675Ala variant was predicted to be deleterious by SIFT, causative of disease by MutationTaster and damaging by Polyphen-2 (Table 2). The PhyloP score was 7.89 and the Grantham distance was 43. This variant was rare in public databases (gnomAD allele frequency 0.0000205). The glutamic acid amino acid affected by the variant was conserved among vertebrates from various species (Figure 1I). 

## 4. Discussion

CHED2 is a heritable ocular disease. Incidences of CHED2 are common in countries in the Global South such as Pakistan [9]. From the reported cases of blindness in Pakistan, CHED2 has a prevalence of 0.9% [9]. PCG is characterized by anomalies in the anterior chamber as well as the trabecular meshwork. Elevated IOP, damaged optic discs and increased corneal diameter are the classical features of PCG [26]. PCG, due to its *CYP1B1* gene variants, is an important cause of blindness in many populations [3]. 

In this study, we explored a large consanguineous family, PKGM3, affected with CHED2, which had been misdiagnosed as PCG. The affected members had been treated with glaucoma surgery for almost 7 years. During 9 years of follow-up clinical evaluation, some specific features related to CHED2, e.g., ground glass-like corneas, were observed in individual IV:9, which is not specific to congenital glaucoma. WES was applied, which revealed a unique case of CHED2 leading to secondary glaucoma. In our study, patients IV:1, IV:2, IV:6, IV:7 and IV: 9 initially underwent repeated surgeries to control glaucoma. Despite the surgical procedures, the patients’ corneas showed gradual opacification. Later on, bilateral penetrating keratoplasties were performed after the CHED2 diagnosis. However, the bilateral penetrating keratoplasties failed to improve the vision. Furthermore, the patients IV:1, IV:2 and IV:6 also had nystagmus (Table 1), which has also been reported previously to coexist with CHED2 [9].

The WES results revealed that all six patients had homozygous c.2024A>C, p.Gly675Ala missense variant in *SLC4A11*. SLC4A11 was originally identified as a bicarbonate transporter but recent studies have shown the role of SLC4A11 in forming a basolateral trans-endothelial cell water pathway [27]. Defects in this protein could give rise to the accumulation of fluid observed in the corneal stroma in CHED patients [27]. Of the 85 mutations associated with CHED2, 50 are missense variants (https://www.hgmd.cf.ac.uk/ac/gene.php?gene=SLC4A11, accessed on 25 August 2022). Most studies reported that they identified variants in all patients screened for the *SLC4A11* gene. However, some studies showed genetic heterogeneity for CHED2, as no variant was reported in the *SLC4A11* gene in some patients [14]. The variant p.Glu675Ala has been previously identified in two Pakistani consanguineous CHED2 families [9]. The reported pedigrees from Pakistan with the same variant had a clear diagnosis with classical signs and symptoms of CHED2 [9]. Clinical assessments showed that the affected members in both families had bilateral swollen and opaque corneas since birth while the age, corneal diameter, axial length, and visual acuity were not mentioned for these affected members [9]. 

CHED2 is sometimes mistakenly diagnosed as PCG. PCG has classical phenotypic features such as an increased IOP, enlargement of the globe (buphthalmos) associated with myopia, Descemet membrane tears, astigmatism, cupping of the optic nerve, and iris atrophy [8]. These signs also appear in developmental glaucoma with juvenile onset. However, these similar phenotypic features can also occur in CHED2 and Posterior Polymorphous Corneal Dystrophy (PPCD), which are sometimes mistakenly diagnosed as early childhood glaucoma and high myopia [20].

A review of the published cases suggested some degree of association between CHED and PCG [7]. Previously, a report described a patient with PCG also suffering from iris hypoplasia along with corneal opacification, who was later shown to have CHED2 [28]. One study reported three cases of glaucoma with CHED [29]. The first case had a familial history of PCG, suffering from ectropion uveae with partial aniridia and nystagmus. In the second case, iris and anterior stroma vascularization were observed. In the third case, the subject had partial aniridia [29]. Another study reported a case of CHED along with glaucoma who also had iris hypoplasia [29]. However, in these previous reports, genetic analysis was not performed; therefore, no causative variant was reported and the mode of inheritance, either recessive or dominant, remained unknown. 

The potential causes of misdiagnosis in our patients could be explained by several possible reasons. CHED2 could be confused with PCG because both diseases may present with opaque corneas, which were clearly observed in this study where all patients showed opaque corneas. Another reason is that the increased central corneal thickness might be associated with a falsely elevated IOP. It has been reported that both CHED2 and PCG lead to corneal edema, which is associated with unreliable measurements of the IOP [7]. As evident in our study, all patients had histories of corneal haziness since birth. Furthermore, examination under anesthesia could also result in an incorrectly measured, high IOP depending upon the nature of the anesthesia agents used (e.g., ketamine could increase IOP) [30]. Even though corneal edema from increased IOP in new-borns also shows buphthalmos [8], buphthalmos with increased IOP is mostly not reported with corneal edema [7]. Buphthalmos is a very important positive sign in early childhood glaucoma; however, non-glaucomatous causes for enlarged eye-globe in early childhood should always be considered [8].

Another reason for falsely measured IOP is the patient cooperation issue [31]. During the examination, the discomfort experienced by the child from a large speculum and the degree of intrathoracic pressure could lead to inaccurate IOP measurements, which cause incorrect diagnoses. In our patients, the individuals were young at initial diagnosis and non-cooperative, which may have led to the initial misdiagnosis of PCG. Finally, a positive family history of PCG could lead to misdiagnosis in some cases [8]. In the present study, a positive family history of PCG led to misdiagnosis in other family members. In these situations, genetic testing is a helpful tool to make differential diagnoses. Although it is possible that PCG and CHED can coexist, it is also possible for CHED to induce congenital glaucoma [8]. However, in the present study, PCG was ruled out after screening all genes associated with congenital glaucoma, and the use of WES led to the diagnosis of CHED2.

Currently, the main treatment option for CHED is corneal transplantation which leads to a risk of infection and graft rejection. Alternative therapies that target the underlying defective transport of water should be further investigated [12]. Clustal Omega alignment showed that glutamic acid at position 675 was evolutionarily conserved among all species (Figure 1I). Bioinformatic tools predicted the variant c.2024A>C (p.Glu675Ala) to be deleterious (Table 2). All the clinical features in patients of the PKGM3 family were severe. In silico analyses demonstrated that the c.2024A>C, p.Glu675Ala mutation can adversely affect protein structure and function. Previous homology modeling and cell culture studies have shown that variant c.2024A>C, p.Glu675Ala impairs the transport of the SLC4A11 protein by changing the translocation pathway which would lead to a catalytically inactive protein [32]. The conserved residue Glu675 throughout all SLC4 family members lies within the catalytic core and was proposed to act as a channel gate [30].

This is the first report of CHED2 with a c.2024A>C, p.Glu675Ala variant as the cause of secondary glaucoma from Pakistan. In the present study, the variant resulted in worse disease progression as compared to those reported before. It is noteworthy that all patients had severe disease progression. Patients IV:1, IV:2, IV6, IV:7 and IV:9 had undergone bilateral trabeculectomies as well as graft rejections. However, no visual improvement was detected. Furthermore, the observation of the c.2024A>C, p.Glu675Ala variant in three different families from Pakistan [9] and the presence of this variant in gnomAD in only individuals from South Asia suggests that it may be a founder mutation in this population.

## 5. Conclusions

In the present study, we suggest that early genetic testing is a powerful tool to avoid misdiagnosis of PCG and CHED2 and to provide timely treatment. We suggest that patients with suspected congenital glaucoma should also be properly screened for positive signs of CHED2. A possible misdiagnosis must be suspected in cases who have corneal edema that fails to clear up even after multiple glaucoma surgeries. Furthermore, examination using ultrasound pachymetry in subjects with PCG may help in differential diagnosis, so a definitive treatment can be undertaken earlier. Our present study signifies the importance of WES genetic testing as a powerful tool to avoid the misdiagnosis of phenotypically similar diseases such as CHED2 and PCG.

## Figures and Tables

**Figure 1 genes-14-00310-f001:**
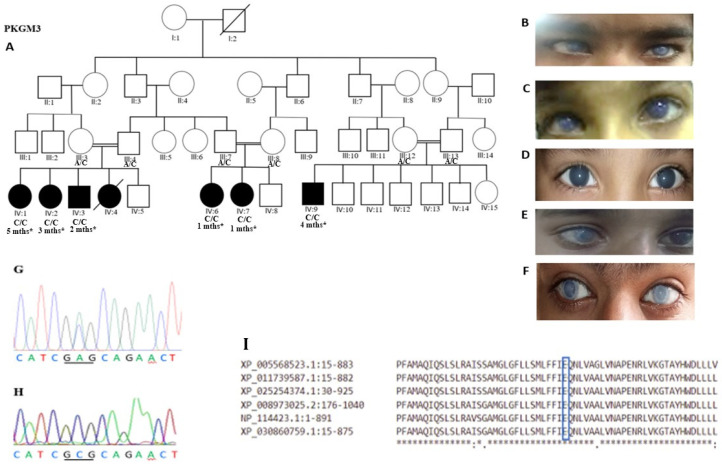
(**A**) Pedigree of the PKGM3 family with their genotypes and age at diagnosis. Squares indicate males and circles indicate females. Black squares and circles indicate individuals with CHED2. White squares and circles are individual without CHED2. Individuals III:3, IV:1 and IV:5 were subjected to exome sequencing. (**B**) Individual IV:1 showing B/L opaque corneas and diffuse corneal edema. (**C**) Individual IV:2 depicting corneal edema and enlarged eye globes. (**D**) Individual IV:3 showing corneal scarring and an enlarged eye. (**E**) Individual IV:7 showing bilateral central corneal scarring along with severe corneal edema. (**F**) Individual IV:9, an advanced case of CHED2 showing bilateral hazy corneas and ground glass appearance. (**G**) Representative electropherogram of *SLC4A11* sequence. Carrier father showing heterozygous sequence c.2024A>C. (**H**) Electropherogram showing homozygous c.2024A>C p.Glu675Ala variant in one affected member. (**I**) Clustal Omega output, showing conservation of p.Glu675 among *SLC4A11* from different species.

**Table 1 genes-14-00310-t001:** Clinical History of CHED2 patients.

Family ID	Individual	Age at Diagnosis	Present Age	Sex	Axial Length OD/OS	CDR OD/OS	IOP (mmHg) OD/OS	Visual Acuity OD/OS	Nystagmus	Corneal Diameter (mm) OD/OS	Surgical Intervention	Clinical Features
**PKGM3**	IV:1	5 mths	16 yrs	F	23.28/23.26	0.4/0.4	20/21	PL/PL	+/+	10/10.5	B/L trab, B/L corneal transplant, B/L graft rejection (2018)	Opaque cornea, diffuse corneal edema, corneal haze
	IV:2	3 mths	8 yrs	F	26.33/25.10	0.3/0.3	18/19	CF/CF	+/+	11.50/11.50	B/L trab, B/L keratoplasty, B/L graft rejection	Corneal haze and cloudy cornea, Descemet membrane fold. B/L flat bleb, enlarged eye globe
	IV:3	2 mths	6 yrs	M	23.5/26.2	0.3/0.4	18/20	CF/CF	−/−	12/11	PKP advised after the age of 10 years	Corneal edema and corneal scarring
	IV:6	1 mth	16yrs	F	26.1/22.1	0.2/0.3	26.1/22.1	HM/CF	+/−	12/11	B/L trab, B/L keratoplasty (no improvement in vision)	Poor Reflex, Corneal haze, photophobia
	IV:7	1mth	14 yrs	F	ND	0.4/0.4	19/19	NPL/NPL	−/−	12/11.5	B/L trab, B/L PKP	Corneal haze, cloudy corena
	IV:9	4 mths	19 yrs	M	25.45/24.53	0.3/0.3	18/19	PL/PL	ND	12/12	B/L trab, B/L corneal transplant	B/L Corneal opacity and haze

**Table 2 genes-14-00310-t002:** Summary of prediction by in-silico tools for the pathogenicity of c.2024A>C, p.Glu675Asp variant.

Prediction Tools	Prediction Result	Severity Score
Mutation taster	Disease-causing	Deleterious
PolyPhen-2	Probably damaging	0.92
PROVEAN	Deleterious	−4.43
I-Mutant	Decrease stability	NA
SIFT	Intolerable	0.98
PhD-SNP	Deleterious	NA
MUpro	Decrease stability	−0.66631328

## Data Availability

All the clinical data has been mentioned in the paper. No supplementary data is needed.

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
