# Peer review of "Exome Sequencing Reveals SLC4A11 Variant Underlying Congenital Hereditary Endothelial Dystrophy (CHED2) Misdiagnosed as Congenital Glaucoma"

_genes, 2023, doi:10.3390/genes14020310_

Round 1

Reviewer 1 Report

The Authors Khazeema Yousaf et.al, have a good article reporting a new variant SLC4A11 determined via exome sequencing leading to secondary glaucoma. Authors suggestion on conducting genome-wide neonatal screening that may avoid misdiagnosis is justified.

Data has been well represented with the corresponding eye images and the IOP levels with other clinical information of CHED2 patients.

It would have been great to learn the existence of SLC4A11 variant in various cohorts of different ethnicity and geographical area.

This report will support the future studies in fine tuning the diagnosis criteria for congenital glaucoma.

Author Response

This reviewer is satisfied with the study and no correction was asked. 

Reviewer 2 Report

While this is an  interesting study where the authors describe a family with CHED where many members were misdiagnosed with PCG, there remain many errors in diagnosis. Also, there are many grammatical errors.

Introduction:

Line 69   ref 13 is wrongly quoted.

Similarly in lines 70 and 71, the authors describe 4 different studies from India , but have quoted only 2.

Results:

Lines 179-183 describe a boy who had been misdiagnosed as PCG but in fact was found to have CHED with secondary glaucoma. It is not clear how the glaucoma was diagnosed in this patient given that his optic disc was healthy. The authors have not mentioned the corneal diameters or axial length to support a diagnosis of Glaucoma.

In fact the title should be : Exome sequencing reveals a SLC4A11 variant underlying Congenital Hereditary Endothelial Dystrophy (CHED2) misdiagnosed as congenital  glaucoma.

Discussion:

Line 276 is incomplete

Line 285 is incorrect as corneal stromal edema may be associated with false low IOP. Maybe the authors can state that increased CCT may be associated with falsely elevated IOP.

The authors should rearrange the discussion such that the clinical and genetic discussion are presented separately. This would provide a better flow to the Discussion. In the present form the genetic discussion begins abruptly in some paragraphs.

Line 318 says that their patient had secondary glaucoma, however in the paragraph above that the authors describe how glaucoma was misdiagnosed in this family. This should be explicitly explained.

The authors should also explain how CHED leads to secondary glaucoma.

Line 332 is grammatically incorrect.
